# Clinical Model for the Prediction of Severe Liver Fibrosis in Adult Patients with Type II Diabetes Mellitus

**DOI:** 10.3390/diagnostics12081829

**Published:** 2022-07-29

**Authors:** Ovidiu Paul Calapod, Andreea Maria Marin, Anca Pantea Stoian, Carmen Fierbinteanu-Braticevici

**Affiliations:** 1Department of Gastroenterology, “Carol Davila” University of Medicine and Pharmacy, 050474 Bucharest, Romania; 2Department of Diabetes, Nutrition and Metabolic Diseases, ”Carol Davila” University of Medicine and Pharmacy, 050474 Bucharest, Romania; 3“Prof. Dr. N. C. Paulescu” National Institute of Diabetes, Nutrition and Metabolic Diseases, 030167 Bucharest, Romania; 4Emergency University Hospital of Bucharest, 050098 Bucharest, Romania

**Keywords:** pSWE, liver fibrosis, diabetes

## Abstract

Background and Objectives: Non-alcoholic fatty liver disease (NAFLD)-related severe liver fibrosis is associated with a higher risk of progressing to decompensated cirrhosis and hepatic failure and developing NAFLD-related hepatocellular carcinoma (HCC), particularly in populations with diabetes. Our pilot study aims to evaluate the performances of various noninvasive methods in predicting liver fibrosis in a population of patients with diabetes and to establish a new scoring system for the prediction of severe fibrosis (>F3). Materials and Methods: A total of 175 patients with diabetes were enrolled for liver fibrosis evaluation. Using the degree of agreement (concordance) between a noninvasive score based on serum biomarkers (NAFLD fibrosis score) and point shear-wave elastography (pSWE) as the reference method, we generated receiver operating characteristic (ROC) curves and performed a multivariate analysis to predict severe liver fibrosis. Results: In our population of patients with diabetes, gamma-glutamyltransferase (GGT), age, body mass index (BMI), the homeostatic model assessment of insulin resistance (HOMA-IR), and glycosylated hemoglobin (HbA1C) were significant predictors for the diagnosis of the F3/F4 group (area under the ROC: 0.767, 0.743, 0.757, 0.772, and 0.7, respectively; *p* < 0.005 for all). Moreover, the combined composite score (the sum of GGT, age, BMI, HOMA index, and HbA1C) had the highest diagnostic performance at a cut-off value of 3 (AUROC—0.899; *p* < 0001). The sensitivity, specificity, negative predictive value (NPV), and positive predictive value (PPV) were 91.20%, 79%, 79%, and 89%, respectively, and 89% of patients were correctly classified as having severe liver fibrosis. In contrast with the Fibrosis 4 (FIB-4) score and the AST-to-platelet ratio index (APRI), the composite score had the best accuracy in discriminating advanced fibrosis. Conclusions: The proposed composite score had a reliable and acceptable diagnostic accuracy in identifying patients with diabetes at risk of having severe fibrosis using readily available laboratory and clinical data.

## 1. Introduction

Non-alcoholic fatty liver disease (NAFLD) includes a wide spectrum of histological entities: non-alcoholic fatty liver (NAFL), non-alcoholic steatohepatitis (NASH), fibrosis, and cirrhosis. Characterized by more than 5% of hepatic fat accumulation and excluding other causes of liver diseases, NAFLD regularly coexists with metabolic disorders, such as type II diabetes mellitus (T2DM) and obesity. A global report on diabetes showed that NAFLD prevalence among the population with diabetes is estimated to be between 70 and 80% [1]. They usually coexist, with the presence of NAFLD increasing the incidence of T2DM and its complications, and T2DM accelerating the progression of NAFLD to more severe forms of liver diseases [2].

Given the central role of the underlying metabolic pathophysiology, an international expert panel consensus proposed a novel redefinition of NAFLD. The metabolic (dysfunction)-associated fatty liver disease (MAFLD) terminology comprises the full spectrum from simple steatosis to stage 4 fibrosis and at least one of the following three criteria: overweight/obesity, the presence of T2DM, or evidence of metabolic dysregulation [3]. This definition better emphasizes the central role of metabolic dysfunctions, and, by requiring a positive diagnosis, MAFLD is no longer only a diagnosis of exclusion. Recent studies have documented the link between high body mass index (BMI) and diabetes via proinflammatory cytokines, insulin resistance, increased levels of circulating fatty acids, and impaired cellular metabolism [4]. Moreover, many large-scale meta-analyses have linked the presence of metabolic syndrome to T2DM due to shared common metabolic pathways, especially insulin resistance and oxidative stress, which are central factors in the pathogenesis of NAFLD [5,6].

As several studies have shown that diabetes is the most common cause of chronic liver disease [2,7], prognostic evaluation and clinical management must promptly determine the status of liver fibrosis in these patients, particularly those with severe fibrosis. This subset of patients carries a higher risk of progressing to decompensated cirrhosis and hepatic failure, developing NAFLD-related hepatocellular carcinoma (HCC), or even death if liver transplantation is not possible [8].

Although the correct diagnosis and the staging of liver fibrosis require a liver biopsy, patients do not frequently accept this. Furthermore, its utility as a diagnostic tool has decreased due to the high prevalence of NAFLD and potentially severe complications [9,10]. Therefore, the potential effectiveness of noninvasive tests that can identify individuals at high risk for disease progression has been investigated in the last decade.

Recently, several ultrasound methods have been refined to enhance the diagnostic accuracy of liver fibrosis [11,12]. Specifically, pSWE, a noninvasive method for liver stiffness measurement (LSM), has great diagnostic efficacy in detecting and staging liver fibrosis in patients with NAFLD, especially in differentiating severe fibrosis (F > 3) [13,14]. Given the well-known overlap of liver stiffness values for mild-to-moderate fibrosis [15], the last update to the consensus on liver elastography by the Society of Radiologists in Ultrasound Liver Elastography recommends that cut-off values greater than 1.7 m/s are highly suggestive of severe fibrosis, especially in patients with viral hepatitis and NAFLD [14]. 

However, several scores based on serum biomarkers have been developed to predict severe liver fibrosis with a high diagnostic performance. Of those noninvasive scores, the NAFLD fibrosis score (NFS), the Fibrosis 4 (FIB-4) score, and the AST-to-platelet ratio index (APRI) have been externally validated in ethnically different populations with NAFLD. [16] Moreover, NFS performs best at distinguishing severe (>F3) from non-severe fibrosis, especially in populations with diabetes, and changes in NFS are linked to mortality [17].

Combining serum biomarkers and elastography methods increases diagnostic performance and is more likely to detect the presence or absence of advanced fibrosis, reducing the need to conduct a liver biopsy [18].

The current pilot study aimed to evaluate the performance of various noninvasive methods for the prediction of liver fibrosis in a population of patients with diabetes, using the concordance of a noninvasive score based on serum biomarkers and pSWE as the reference method. In particular, we focused on severe fibrosis, as this is the key determinant of the disease prognosis, prioritization for treatment, and even establishing the potential for reversibility [19,20]. Therefore, based on these results, we established a new scoring system for the prediction of severe fibrosis in patients with diabetes.

## 2. Materials and Methods

### 2.1. Study Population

A prospective study was carried out in the Bucharest Emergency University Hospital, Romania, following approval by the Ethics Committee of the Emergency University Hospital of Bucharest (no. 9195/17 February 2021). The inclusion criteria were as follows: adult patients (>18 years of age) with established T2DM for more than 6 months, which was defined using the criteria of the American Diabetes Association (ADA) “Standards of Medical Care in Diabetes” with the standard of care treatment [21]. All patients underwent an in-depth medical screening, which included screening for secondary causes of liver disease based on medical history, a physical examination, and biochemical markers. Eligible patients were negative for anti-hepatitis C virus antibodies and hepatitis B surface antigens. Moreover, patients with other etiologies of liver diseases, including autoimmune hepatitis, drug-related hepatitis, cardiac failure or significant valvular disease, active malignancy, Wilson’s disease, and hemochromatosis, were excluded. Similarly, screening for alcohol consumption was conducted using a questionnaire adopted by the World Health Organization, and females with an alcohol consumption of >20 g/day and males with an alcohol consumption of >30 g/day were excluded from the analysis (alcohol use disorders identification test) [22]. We enrolled 175 consecutive patients with diabetes undergoing liver steatosis and fibrosis evaluation in the Gastroenterology Department, of whom 24 were excluded due to unreliable pSWE measurements, and 17 more were excluded due to inconsistency between pSWE and NFS. Before the examination, written informed consent was obtained from all the patients, and the Ethical Principles for Medical Research Involving Human Subjects in the World Medical Association Declaration of Helsinki were complied with. 

### 2.2. Anthropometry and Risk Factor Measurements

Demographic data, gender, and age were collected for each subject. Body weight was assessed using a calibrated scale (Seca, Germany, model: 701). Body height was recorded using a non-stretchable measurement tape. BMI was defined as body weight (kilograms) divided by the square of body height (m^2^). Waist circumference was measured at the superior border of the iliac crest with a flexible non-stretchable measuring tape. Arterial hypertension was diagnosed according to the “2020 International Society of Hypertension Global Hypertension Practice Guidelines” [23].

### 2.3. Laboratory Tests and Metabolic Syndrome

After a 12 h overnight fast, blood tests were collected for all patients, including liver function tests (aspartate amino transferase (AST); alanine amino transferase (ALT); gamma-glutamyltransferase (GGT), albumin, and the international normalized ratio (INR)), total cholesterol, low-density lipoprotein cholesterol (LDL-C), high-density lipoprotein cholesterol (HDL-C), triglycerides, fasting glucose, glycated hemoglobin (HbA1c), and fasting insulin. A homeostatic model assessment of insulin resistance (HOMA-IR) was calculated using the following score HOMA-IR = (Fasting insulin) * (Fasting glucose)/405 [24].

To assess the presence of metabolic syndrome (MS), we started from the assumption that a patient must have three or more of the following metabolic disarrangements: an elevated fasting glucose of l00 mg/dL or greater, a waist circumference more than 102 cm in men and 88 cm in women, blood pressure values of systolic 130 mmHg or higher and/or diastolic 85 mmHg or higher, reduced high-density lipoprotein cholesterol (HDL) less than 40 mg/dL in men and less than 50 mg/dL in women, and elevated triglycerides of 150 milligrams per deciliter of blood (mg/dL) or greater [25].

### 2.4. Liver Fibrosis Assessment Using Point Shear-Wave Elastography 

A standard B-mode ultrasound assessment was performed to subjectively assess the degree of liver steatosis. Mild steatosis (Score 1-S1) was defined as a slight increase in hepatic echogenicity with a standard diaphragm visualization, moderate steatosis (Score 2-S2) was defined as a moderate increase in hepatic echogenicity with a moderately impaired diaphragm visualization, and severe liver steatosis (Score 3-S3) was defined as a marked increase in hepatic echogenicity and an inability to visualize the diaphragm [26].

All patients underwent pSWE using Siemens Acuson S2000, version VB20, Model no. 10041461 (Siemens Healthineers, 91052 Erlangen, Germany), with a 4C1 transducer (4 MHz). The Virtual Touch Tissue Quantification mode (Siemens Medical Solutions, Mountain View, CA, USA) of the ultrasound machine was used to assess the LSM. The same gastroenterologist, with more than five years of pSWE experience, performed the procedure. The overall assessment was carried out in accordance with the latest updates published by the Society of radiologists in ultrasound liver elastography [14]. The measurement was taken through an intercostal space in segments 6 and 8, with the patient in the supine position with their right arm abducted, during a breath hold. In all individuals, pSWE was performed at least 15–20 mm below the liver capsule for high accuracy. A total of 10 valid measurements were taken, and the results are expressed as a median. The technical quality parameters were interquartile range intervals (IQRs) (defined as the difference between the 75th and 25th percentiles) and success rates (SRs) (the number of successful measurements divided by the total number of measurements). In addition, an IQR/median (IQR/M) ratio was calculated, and a reliable acquisition was considered only if IQR/M was ≤15% and the SR was at least 60%. Unreliable measurements were defined as the impossibility of obtaining ten valid measurements or proper quality parameters (IQR ≤ 30% and/or SR ≥ 60%). In agreement with actual recommendations, we used 1.7 m/s as a cut-off value for severe fibrosis [14,15]. 

### 2.5. Liver Fibrosis Assessment Using Serum Markers

According to published algorithms [27], several liver fibrosis scores, such as the NAFLD fibrosis score (NFS), the Fibrosis 4 (FIB-4) score, and the AST-to-platelet ratio index (APRI), were calculated for each patient.

### 2.6. Liver Fibrosis Definition

In agreement with the latest recommendations [18], we used a combined method to diagnose advanced fibrosis in our population: the concordance between the NFS score (>0.65) and the pSWE values (>1.7 m/s), or the proportion of patients for whom both tests yielded the same results. We distinguished two groups of patients: the F1/F2 group, defined as having both an NFS score < 0.65 and a pSWE value < 1.7 m/s, and the F3/F4 group, defined as having both an NFS score > 0.65 and a pSWE value > 1.7 m/s.

### 2.7. Statistical Analysis

The data were stored in Microsoft Office Excel 2019 version. Statistical analyses were performed using Epi Info version 7.2.4.2020 and IBM-SPSS software version 20.0.0. The distribution of variables was evaluated using the Kolmogorov–Smirnov test. Continuous variables with a normal distribution are presented as means ± standard deviations (SDs). Data without a normal distribution are presented as medians and interquartile ranges, and categorical variables are presented as percentages. To detect the presence of a statistically significant difference between groups, we performed Student’s *t* test and the Mann–Whitney U test, and *p* < 0.05 was considered significant. A receiver operating characteristic (ROC) curve analysis was used to identify the cut-off values for the prediction of severe fibrosis. The sensitivity, specificity, positive predictive values (PPVs), and negative predictive values (NPVs) were also calculated. We considered a confidence level of 95% for the estimation of intervals. Multivariate logistic regression analyses were used to determine the parameters related to severe fibrosis. We established a new scoring system—a composite score—for the prediction of severe fibrosis based on these parameters. The area under the ROC (AUROC) was used to assess the diagnostic performances of the noninvasive tests.

## 3. Results

Of the 175 patients with known T2DM who were evaluated for liver fibrosis in the Gastroenterology Department, 24 (13.7%) were excluded due to unreliable pSWE measurements. Compared to the reliable group (n—151), the unreliable group was older (62.3 ± 12.8 vs. 51.6 ± 13.5, *p*—0.874) and mostly comprised men (47.5% vs. 45.1%, *p*—0.478), but this was not statistically significant. The BMI of the unreliable group was higher (32.4 ± 2.8 vs. 29.3 ± 2.5, *p*—0.147), but the difference did not reach statistical significance.

The rate of agreement between the NFS score and the pSWE values, or the proportion of patients for whom both tests yielded the same results, was met in 134 patients (88.7%). These patients were recruited for the final evaluation (Table 1). Based on these values, the patients were divided into two groups (the F1/F2 group, defined as having both an NFS score < 0.65 and a pSWE value < 1.7 m/s, and the F3/F4 group, defined as having both an NFS score > 0.65 and a pSWE value > 1.7 m/s). A comparison of the clinical and biochemical characteristics of the two groups of patients is shown in Table 2.

The prevalence of liver steatosis in our study group was 85.8% (n—115). We found that 37.3% (n—50) of patients had S1 steatosis, 44% (n—59) had S2 steatosis, and 17.9% (n—25) had S3 steatosis.

The prevalence of severe fibrosis (F3/F4) based on pSWE and NFS scores was 18.7% (n—25), while 81.3% (n—109) of patients were classified as not having severe fibrosis (F1/F2). Patients with severe fibrosis were more likely to be older and have a higher waist circumference. Furthermore, they were more likely to have metabolic syndrome and low levels of albumin. Moreover, patients with severe fibrosis had greater levels of serum triglycerides, AST, ALT, GGT, serum glucose, and HbA1c, and a higher HOMA-IR. Regarding the noninvasive markers of liver fibrosis, patients with severe fibrosis had a statistically significant increase in FIB-4, APRI, and NFS scores.

Table 3 shows the diagnostic performances and accuracies of the parameters predicting the presence of F3/F4 in patients with T2DM. GGT, age, BMI, HOMA index, FIB-4, APRI, NFS, and HbA1C were significant predictors for the diagnosis of the F3/F4 group.

Univariate and multivariate analyses were performed, and the results are displayed in Table 4. Using the cut-off points obtained from ROC analyses, GGT (>113 U/L), age (>55 years), BMI (>30.1 kg/m^2^), HOMA-IR (>3.3), and HbA1 c (>6.5%) were independently associated with the prediction of the F3/F4 group.

Based on the results of the multivariate analyses, we created a new scoring system that comprises the following parameters: GGT values of <113 U/L and ≥113 U/L were scored as 0 and 1, respectively; ages < 55 years and ≥55 years were scored as 0 and 1, respectively; BMIs of <30.1 and ≥30.1 were scored as 0 and 1, respectively; HOMA-IR values of <3.3 and ≥3.3 were scored as 0 and 1, respectively; and HbA1c levels of <6.5% and ≥6.5% were scored as 0 and 1, respectively. The composite score was defined as the sum of the GGT value, age, BMI, HOMA value, and HbA1C level. The receiver operating characteristic (ROC) curves for the prediction of F3/F4 are shown in Figure 1 and Figure 2. The composite score had the highest significant diagnostic performance for the prediction of F3/F4 (0.899(0.792–0.986), *p* < 0.005), followed by NFS, APRI, and FIB-4 scores. Using a cut-off score of 3 points, the sensitivity, specificity, PPV, and NPV were 85.3%, 91.2%, 79%, and 89%, respectively.

## 4. Discussion

Our study proposes a new composite scoring system, named the composite score, for the prediction of severe fibrosis in patients with T2DM. Through multivariable logistic regression analyses, we showed that GGT values, age, BMI, HbA1C, and HOMA-IR were the strongest predictors of advanced fibrosis in patients with T2DM. Regarding BMI, HbA1C, and HOMA-IR, the association of severe fibrosis with metabolic syndrome is well documented and has a strong correlation [28,29]. Moreover, the higher the number of metabolic dysregulations in a patient, the greater the severity of the fibrosis [28]. Prior studies have used similar clinical and serum biomarkers to predict the presence or absence of severe fibrosis in individuals with NAFLD. In addition, some authors have developed a simple clinical model with a moderate diagnostic performance for the prediction of severe fibrosis using liver enzymes and diabetes status [30].

Another example is the Fatty Liver Index, which uses triglyceride level and waist circumference to predict NAFLD [31]. Bazick et al. collected data from a large cohort derived from the NASH Clinical Research Network studies and demonstrated that advanced age, higher AST/ALT and waist-to-hip ratios, hypertension, isolated abnormal alkaline phosphatase, hematocrit, serum insulin, and low platelet counts are associated with advanced liver fibrosis [32]. Another study found an association between GGT values and increased mortality in advanced fibrosis [33]. Unlike prior research, our study focuses on patients with T2DM, a population known to have a higher risk of severe fibrosis and mortality [2,34]. Moreover, no significant predictive values were found in our study for liver function tests, waist circumference, platelets, or triglycerides. Our model performed better than the FIB-4 and APRI indexes. A large meta-analysis evaluating the diagnostic performance for the prediction of severe fibrosis in NAFLD of the FIB-4 and APRI scores reported AUROCs of 0.81 (0.73–0.89) and 0.82 (0.74–0.89), respectively, results that are consistent with ours [35]. 

Other combinations of serum biomarkers for liver fibrosis, which involve the measurement of markers of matrix turnover, have been analyzed in individuals with NAFLD [15]. These noninvasive tests, such as The Enhanced Liver Fibrosis test, have been evaluated in large cohorts of patients with NAFLD and performed only marginally better than our clinical model in predicting severe fibrosis (AUROC 0.91 vs. 0.899) [15]. However, these tests are relatively expensive, and they are not widely available.

A recent meta-analysis that evaluated the diagnostic accuracy of pSWE in patients with NAFLD showed that, in the staging severe fibrosis (F > 3), the pooled sensitivity and specificity were 0.92 (0.87, 0.95) and 0.85 (0.80, 0.89), respectively, with an AUROC of 0.94 (0.92, 0.96) [13]. As the cut-off values for staging hepatic fibrosis vary across studies and different vendors, several studies contributed data and values to improve clinical decisions [36,37,38]. Given the well-known overlap of liver stiffness values for mild-to-moderate fibrosis [15], the last update to the consensus on liver elastography by the Society of Radiologists in Ultrasound Liver Elastography states that cut-off values greater than 1.7 m/s are highly suggestive of severe fibrosis [14]. In our study, we reported an unreliable measurement rate in our population that is higher than that in other studies [38,39], especially in older patients with a higher BMI. As prior studies state, one possible hypothesis for this trend is that individuals with higher BMIs have a large quantity of adipose tissue between the skin and the liver capsule [40].

Of the 151 patients with diabetes for whom we obtained reliable measurements, the concordance between the NFS scores and the pSWE values was met in 134 patients (88.7%), with a prevalence of severe fibrosis (F3/F4) of 18.7% (n—25), while 81.3% (n—109) of patients were classified as not having severe fibrosis (F1/F2). Previous studies that used liver biopsies as the gold standard have shown that the percentage of severe fibrosis among patients with T2DM ranges from 16.2% to 43.1% [41,42]. Bharat K Puchakayala et al., who conducted a study that assessed the clinical and histopathological features of NAFLD, found that the prevalence of severe fibrosis was significantly higher in patients with T2DM than in those without T2DM (27.3 % vs. 13.3 %, *p* < 0.01) [43]. Moreover, an extensive meta-analytic systematic review of seven studies published between 2004 and 2017, which included 439 liver biopsies from patients with T2DM, showed that the mean reported severe fibrosis prevalence was 22.01% (ranging from 3.3% to 50%) [44].

The clinical profile in our study is consistent with data from more extensive studies [45,46]. We found that BMI, waist circumference, serum triglycerides, AST, ALT, GGT, serum glucose, HbA1C, and HOMA-IR were significantly higher in patients with severe fibrosis. Moreover, these patients frequently exhibited metabolic syndrome. In light of this data, our study supports the notion that the prevalence of severe fibrosis, the main determinant of mortality in NAFLD, is higher in patients with metabolic alterations. Other studies found a bidirectional correlation between liver fibrosis and peripheral diabetic neuropathy, suggesting possible common etiological mechanisms [47,48,49].

When looking at the independent effect of each studied parameter on severe fibrosis, multivariate logistic regression identified that GGT values, age, BMI, HbA1C, and HOMA-IR were significantly associated with it. We found no statistically significant independent association with arterial hypertension, but in a large meta-analysis, Singh et al. established that elevated blood pressure is a strong risk factor for rapid progression to severe fibrosis and cirrhosis [50]. The assessed outcomes could explain this difference; in our study, we evaluated the risk factors associated with the presence of severe fibrosis, while in Sing’s meta-analysis, the primary outcome was the estimation of the fibrosis progression rate. 

The clinical score created in this research can help identify severe liver fibrosis in patients with T2DM and support clinicians in deciding whether to send patients for a liver biopsy [51]. However, according to this score, clinical judgment, including liver biopsies, may still be needed for patients who are not found to have severe fibrosis. Nevertheless, this score would correctly classify 89% of patients with advanced fibrosis.

The present study has several strengths. Our cohort was consecutively enrolled and described using clinical, laboratory, and pSWE parameters. In particular, we focused on documenting diabetes status (the duration of T2DM, serum glucose, HbA1c, HOMA-IR, and insulin use), as this is a well-studied risk factor for advanced liver fibrosis. To the best of our knowledge, our study is one of few that evaluates the clinical, laboratory, and pSWE parameters of NAFLD and the links between them in a population at risk, such as those with T2DM. 

However, there are some limitations to consider. This is a pilot study, conducted on a small study population, and more significant large-scale quantitative analyses are needed for validation and to assess the technical challenges of pSWE. As liver biopsy indication has decreased and the use of serum biomarkers and elastography imaging has become more frequent in recent years, conducting a liver biopsy was not feasible in our patients. In addition, our cohort study included a population of Caucasians, who showed a high prevalence of obesity and severe fibrosis. These results need to be validated on different populations, as pSWE can be a non-reliable technique in patients with high BMIs. Another limitation concerns pSWE cut-off values. The diagnostic accuracy of these values for liver fibrosis staging varies across different ultrasound systems and liver fibrosis etiologies. Moreover, there are no standardized cut-offs for pSWE in patients with NAFLD, and the value of 1.7 m/s for discriminating F ≥ 3 in NAFLD has insufficient evidence. A recent systematic review and meta-analysis, which included 29 diagnostic studies, found a cut-off value for severe fibrosis that ranges from 1.34 to 2.20 m/s for NAFLD and NASH, a relatively large range, which may be the main determinant for the high heterogeneity between reports [10]. Thus, the cut-off values need further rigorous studies for confirmation in populations with NAFLD.

## 5. Conclusions

In conclusion, in patients with T2DM, GGT values, age, BMI, HbA1C, and HOMA-IR were the strongest predictors of advanced fibrosis. Using readily available laboratory and clinical data, the proposed composite score had reliable and acceptable diagnostic accuracy in identifying patients with diabetes at risk of having severe fibrosis. This may help clinicians decide who should be referred to a hepatologist or for a liver biopsy for further evaluation. Further multicentric, large-scale studies and validation cohorts are needed to validate this score as a screening test.

## Figures and Tables

**Figure 1 diagnostics-12-01829-f001:**
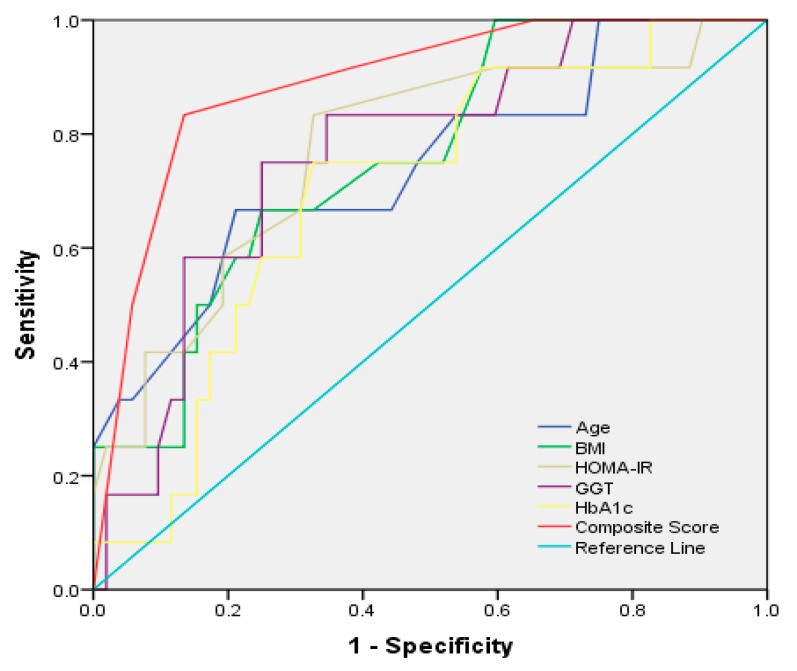
Receiver operating characteristic (ROC) curves for GGT, AGE, BMI, HOMA-IR, HbA1c, and composite score for the detection of F3/F4 group of patients.

**Figure 2 diagnostics-12-01829-f002:**
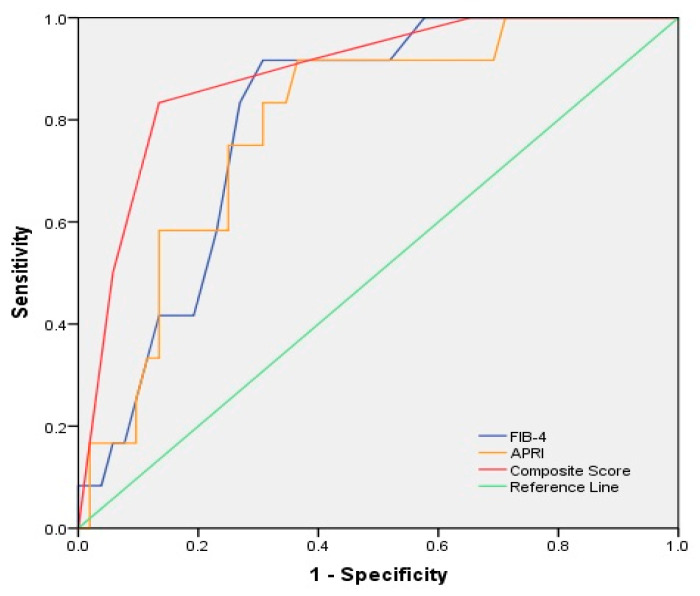
Receiver operating characteristic (ROC) curves for FIB-4, APRI, and composite score for the detection of F3/F4 group of patients.

**Table 1 diagnostics-12-01829-t001:** Baseline characteristics of the 134 patients with diabetes.

Parameters	Value
Female (%)	56.25
Male (%)	43.75
Age (years)	49.39 ± 8.19
Waist circumference (cm)	92.28 ± 10.08
BMI (kg/m^2^)	29.33 ± 2.51
Hypertension (%)	73.85
MS (%)	74
Cholesterol (mg/dL)	227.42 ± 65.2
HDL-C (mg/dL)	36.21 ± 6.05
LDL-C (mg/dL)	106.3 ± 47.2
Serum triglycerides (mg/dL)	195.65 ± 58.91
AST (U/L)	83.98 ± 18.11
ALT (U/L)	86.41 ± 30.41
GGT (U/L)	107.02 ± 52.17
Albumin (g/dL)	3.7 ± 0.58
INR	1.03 ± 0.14
Duration of T2DM (years)	7.7 (3.4–11.5)
Insulin use (%)	62.25
Serum glucose (g/dL)	114.45 ± 19.88
HbA1c (%)	6.55 (5.1–9.2)
HOMA-IR	3.06 ± 0.54
NFS	0.29 (−1.5–2.4)
FIB-4	2.74 (0.66–9.41)
APRI	1.02 ± 0.43
pSWE (m/s)	1.3 ± 0.45

BMI—body mass index; MS—metabolic syndrome; HDL-C—high-density cholesterol; LDL-C—low-density cholesterol; AST—aspartate amino transferase; ALT—alanine amino transferase; GGT—glutamyltransferase; INR—international normalized ratio; HbA1c—glycosylated hemoglobin; HOMA-IR—homeostatic model assessment of insulin resistance; NFS—NAFLD fibrosis score; FIB-4—Fibrosis 4; APRI—AST-to-platelet ratio index; pSWE—point shear-wave elastography.

**Table 2 diagnostics-12-01829-t002:** Comparison of clinical and biochemical parameters.

Variables	F1/F2	F3/F4	*p*-Value *
Sex			
Female (%)	48.43	7.82	0.021
Male (%)	32.82	10.93	0.032
Age (years)	48.66 ± 8.19	55.41 ± 6.85	<0.001
Waist circumference (cm)	91.96 ± 9.51	94.66 ± 12.64	0.053
BMI (kg/m^2^)	28.11 ± 2.45	30.22 ± 2.77	0.02
Hypertension (%)	78.3	69.4	0.087
MS (%)	68.30	79.7	<0.001
Cholesterol (mg/dl)	228.76 ± 67.75	221.6 ± 54.92	0.473
HDL-C (mg/dl)	37 ± 6.03	32 ± 5.09	0.341
LDL-C (mg/dl)	105.4 ± 45.6	109.2 ± 48.5	0.654
Serum triglycerides (mg/dl)	194.49 ± 60.7	202.58 ± 54.92	0.032
AST (U/L)	82.15 ± 18.14	91.94 ± 16.3	0.021
ALT (U/L)	85.09 ± 31.9	92.14 ± 23.13	0.047
GGT (U/L)	99.36 ± 48.76	140.02 ± 57.58	<0.001
Albumin (g/dl)	3.75 ± 0.56	3.46 ± 0.59	0.054
INR	1.03 ± 0.14	0.14 ± 0.12	0.847
Duration of T2DM (years)	7.1 ± 3.4	8.4 ± 4.1	0.547
Insulin use (%)	64.2	60.3	0.352
Serum glucose (g/dl)	112.21 ± 18.66	124.18 ± 22.83	0.046
HbA1c (%)	6.2(5.1–8.2)	7.1(5.9–9.2)	0.032
HOMA-IR	2.99 ± 0.42	3.38 ± 0.85	0.025
NFS	0.01 (−1.5–1.8)	1.48 (0.74–2.4)	<0.001
FIB-4	2.3 ± 0.74	4.68 ± 1.95	0.021
APRI	0.87 ± 0.2	1.69 ± 0.5	0.015
pSWE (m/s)	0.87 ± 0.22	2.04 ± 0.26	<0.001

* Values are statistically significant at *p* < 0.05; BMI—body mass index; MS—metabolic syndrome; HDL-C—high-density cholesterol; LDL-C—low-density cholesterol; AST—aspartate amino transferase; ALT—alanine amino transferase; GGT—gamma-glutamyltransferase; INR—international normalized ratio; HbA1c—glycosylated hemoglobin; HOMA-IR—homeostatic model assessment of insulin resistance; NFS—NAFLD fibrosis score; FIB-4—Fibrosis 4; APRI—AST-to-platelet ratio index; pSWE—point shear-wave elastography.

**Table 3 diagnostics-12-01829-t003:** Diagnostic performances of different parameters in predicting severe liver fibrosis.

Variables	Cut-Off Value	AUROC (95% CI)	Std. Error	Sensibility (95% CI)	Specificity (95%CI)	PPV (95% CI)	NPV (95%CI)	*p*-Value *
GGT, U/L	113	0.76 (0.627–0.907)	0.72	75.00%	75.00%	88.00%	72.00%	0.004
				(62.7–87.3%)	(61.3–85.5%)	(67.7–97.1%)	(60.6–82.3%)	
Age, years	55	0.74 (0.578–0.908)	0.84	58.30%	80.80%	89.00%	61.30%	0.009
				(40.2–71.4%)	(62.6–95.8%)	(68.4–98.3%)	(45.6–76.1%)	
BMI, kg/m^2^	30.1	0.75 (0.617–0.897)	0.71	67.60%	75.00%	93.00%	63.40%	0.006
				(49.2–78.8%)	(60.9–85.6%)	(78.4–97.9%)	(46.2–77.7%	
HOMA-IR,	3.3	0.77 (0.616–0.924)	0.78	66.70%	88.80%	95.00%	78.00%	0.004
				(48.7–79.4%)	(66.3–97.1%)	(81.2–99.3%)	(65.4–90.9%)	
HbA1c, %	6.5	0.70 (0.548–0.852)	0.78	64.30%	72.20%	90.50%	61.20%	0.032
				(48.3–80.5%)	(59.8–82.6%)	(69.7–99.3%)	(58.9–75.4%)	
FIB-4	1.35	0.80 (0.689–0.915)	0.58	82.3%	65.2%	74%	92.1%	<0.001
				(62.9–96.6%)	(48.3–78.8%)	(60.7–82.9%)	(78.5–99.9%)	
APRI	1	0.79 (0.663–0.918)	0.65	72.4%	87%	72.3%	89.6%	0.002
				(58.4–81.9%)	(66.1–97.3%)	(57.9–80.1%)	(67.8–98.1%)	
NFS	>0.65	0.82 (0.67–0.93)	0.63	78.4%	63.2%	67.4%	94.3%	
				(61.2–89.2%)	(50.1–79.5%)	(52.8–81.3%)	(79.4–99.3%)	<0.001

* Values are statistically significant at *p* < 0.05; BMI—body mass index; GGT—gamma-glutamyltransferase; HbA1C—glycosylated hemoglobin; HOMA-IR—homeostatic model assessment of insulin resistance; FIB-4—Fibrosis 4; APRI—AST-to-platelet ratio index.

**Table 4 diagnostics-12-01829-t004:** Univariate and multivariate analyses for factors related to F3/F4 group.

		Univariate Analyses		Multivariate Analyses	
Variables	Cut-Off Values	Or (CI 95%)	*p*-Value *	OR (CI 95%)	*p*-Value *
GGT, U/L	≥113	1.912 (1.534–2.861)	0.0479	8.993 (2.11–38.311)	0.003
BMI, kg/m^2^	≥30.1	1.544 (1.121–2.128)	0.0079	5.996 (1.548–23.245)	0.009
Age, years	≥55	1.347 (1.0376–1.589)	0.0074	7.453 (1.889–29.401)	0.004
HOMA-IR	≥3.3	3.342 (1.874–8.457)	0.0271	5.879 (1.5413–22.431)	0.009
HbA1c, %	≥6.5	2.6616 (1.7613–3.446)	0.0051	6.851 (1.954–19.547)	0.002

* Values are statistically significant at *p* < 0.05; BMI—body mass index; GGT—gamma-glutamyltransferase; HbA1c—glycosylated hemoglobin; HOMA-IR—homeostatic model assessment of insulin resistance.

## Data Availability

Data are available upon request from the authors.

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
