# Peer review of "Clinical Model for the Prediction of Severe Liver Fibrosis in Adult Patients with Type II Diabetes Mellitus"

_diagnostics, 2022, doi:10.3390/diagnostics12081829_

Round 1
Reviewer 1 Report
Calapod et al have written an interesting and valuable original research article in which they proposed a clinical model which allows for predicting severe liver fibrosis in adult patients diagnosed with type 2 diabetes mellitus. Obesity, metabolic syndrome, its components, and complications. This is a vital subject because obesity and its comorbidities are considered to be a contemporary epidemic and one of the most important public health problems worldwide. Although the manuscript is generally well prepared and should be considered for publication in Diagnostics, I recommend some modifications that may further improve the quality of the paper.
The Authors presented clearly in the introduction the most important information about NAFLD and tools useful in the assessment of liver fibrosis. Attention was also paid to the relationship between NAFLD and type 2 diabetes mellitus. In my opinion, it is worth mentioning that diabetes mellitus may occur not separately, but in a lot of cases, it coexists with other components of metabolic syndrome. Moreover, it is worth noting about the role of oxidative stress in the pathogenesis of NAFLD and oxidative stress may be considered a link between type 2 DM and NAFLD. As mentioned, DM type 2 often (although not always) coexists with other components of metabolic syndrome and occurs in obese patients, similarly to NAFLD. Significant differences were found depending on body mass and coexisting metabolic disturbances. Moreover, “obesity and insulin resistance” was shown to be the most important component of metabolic syndrome taking into account its relationship to oxidative stress. (See, for example, the following references: doi.org/10.1155/2021/9987352; doi.org/10.3390/antiox11010079; doi.org/10.1155/2021/9986299; doi.org/10.3390/cells9041041; doi.org/10.3390/biom10121702).
In chapter 2.3. the expression “liver function tests” should be explained, what exactly laboratory tests do the Authors mean. Pay attention, please, that INR is mentioned in a further part of the same sentence, although elevated INR is one of the most sensitive parameters which is associated with liver insufficiency.
Abbreviations from high-density and low-density lipoprotein cholesterol should be HDL-C and LDL-C, respectively, not HDL-c and LDL-c, in my opinion. On the other hand, the abbreviation for “glycated hemoglobin” should be HbA1c, not HbA1C. Do not use the expression “glycosylated hemoglobin” but “glycated hemoglobin”, please, because glycation is a process without the participation of any enzyme, whereas glycosylated is a process enzymatically catalyzed.
Please explain how you tested the consistency of quantitative variables with the normal distribution. In the case of variables with a distribution consistent with the normal distribution, please present descriptive statistics in the form of a mean and standard deviation, and in the case of variables with a distribution significantly different from the normal distribution, the optimal way of presenting descriptive statistics seems to be the median and the interquartile range.
Differences between patients with and without severe fibrosis may be presented in a table (I mean information from lines 207-219). It would make such information better visible and readable.
Effects of univariate and multivariate analysis (Table 4) should be presented in two separate tables.
Consider, please, present this study as a pilot study, because the population is really small when we talk about a study, in which a new diagnostic model was presented and its predictive value was evaluated.
The list of references should be prepared according to MDPI instructions.
Author Response
I am grateful for your insightful comments on our paper. Thank you for the time and effort that you have dedicated to providing your valuable feedback on our manuscript. You will find all our answers to your suggestions in the attached file.
Kind Regards,
Ovidiu Calapod

Reviewer 2 Report
The aim of this study was to evaluate the performance of various noninvasive biomarkers for predicting liver fibrosis in a series of diabetic patients and to establish a new scoring system for predicting severe fibrosis.
The study is interesting; there are some inaccuracies that need to corrected.
The discussion must be shortened, focused on discussing the results of the study and rearranged highlighting first the main results, i.e., the composite score and its performance.
Other comments:
Line 15: do you mean study instead of “paper”?
Lines 17-20: this sentence in unclear. Please reword.
Line 28: which were these other noninvasive scores? Please give details.
Line 63: pSWE is not an “emerging” noninvasive method. It’s a well-established technique available since more than a decade.
Line 64: pSWE doesn’t give an image. The word “imaging” must be deleted. By the way, the term “ARFI imaging” identifies a technique similar to strain elastography, therefore it’s incorrect to use it for pSWE.
Line 67-68: this sentence is incorrect. The Society of Radiologists in Ultrasound (SRU) released an update to the consensus on liver elastography. Please reword here and thereafter (lines 145-146; line 276).
Line 126: this statement is incorrect: The patient should have three or more of these criteria and not “any three”. Please reword and check that the patients were classified correctly following these criteria.
Line 140: the company is Siemens Healthineers, and it is based in Germany. Please correct.
Line 154: the update to SRU consensus suggests an IQR/M up to 15% because the conversion between m/sec and kPa is not linear.
Line 177: 95% confidence intervals must be calculated for sensitivity, specificity, PPV and NPV and shown in the results.
Lines 209-219: the numbers already given in the table must not be duplicated in the text. This list is confusing and doesn’t add any additional information. All these numbers must be deleted.
Lines 223-228 (Table 3): Give just the title of the table. All the numbers are already present in the table. They must not appear in the title. Two decimal digits for the AUROCs are enough.
Lines 229-21: the statistical significance is already given in the table. There isn’t any need to duplicate this result.
Lines 235-236: Again, no need to duplicate the results. Please delete.
Lines 258-268: This is the background of the study. Please delete these sentences and discuss the result of the study instead. The discussion must start highlighting the main result, i.e., the composite score.
Line 308-318: I’d suggest to move this part to the beginning of the discussion.
Author Response

(The authors gave the same response as above.)

Reviewer 3 Report
In the present study the authors propose a new score (Composite Score) based on clinical and biochemical criteria to evaluate liver fibrosis in patients with type II diabetes. This score is based on GGT, age, BMI, HOMA index and HbA1C: each parameter has a threshold value. GGT ≥ 113 U/l, age ≥ 55, BMI ≥ 30.1, HOMA-IR ≥ 3.3, HbA1C ≥ 6.5% are scored 1 point each. Using a cut-off of 3 points, this Score has a significant diagnostic performance for the prediction of severe liver fibrosis.
We have the following concerns:
- My main concern is related to the real value of the paper; indeed, the authors used as a gold standard NFS, which includes variables largely overlapping with those of the composite score. What is the additional value of using the composite score?
- A major limitation, as the authors admit, is the small sample size: the severe fibrosis group accounts only 25 patients
- There is no evaluation of normal distribution; indeed, it is reasonable that the distribution of continuous variables in the present population is not normal. Thus medians [IQR] and Mann-Whitney test should be preferred
- The results section is redundant. For instance, there is no reason to repeat extensively the results of table 2 in the text;
- Table 3 is not introduced in the text
- It is unclear how the authors selected the variables included in the composite score. For instance they excluded categorical variables; moreover, among the continuous ones, some were not included (like AST/ALT)
- Table 4 is absolutely unclear; they joined together univariate and multivariate analysis in a single table, making really difficult to understand the analysis performed. Once more, there is no reason to extensively repeat the results in the table and in the text;
- The authors added in Table 3 and in Table 4 the composite score before introducing it. I suggest to exclude it from multivariate analysis and from the tables; to introduce it after the multivariate analysis, adding in the text its diagnostic performances.
- The authors compared the ROC curves in Fig.1 and 2. In Fig.1 they included composite score and other clinical variables; in fig.2 composite score was compared with other scores. In the text the authors stated that composite score performed better, buti t is unclear whether the p scores refers to the comparison reported in fig.1 or in fig.2.
- In paragraph 2.4 “Liver fibrosis assesment by point shear wave elastography”, the measurement of steatosis by standard-B mode ultrasound is mentioned. However, in the results section there is no mention of such parameter.
- Finally, I suggest a revision of written English.
- All the abbreviations must be clarified at their first appearance (for instance, this is not the case of ARFI, pSWE)
Author Response

(The authors gave the same response as above.)

Reviewer 4 Report
The authors investigated the possibility of multiple noninvasive parameters for predicting severe liver fibrosis (F3/F4) in NAFLD patients with T2DM and created a new composite scoring system (the sum of GGT, age, BMI, HOMA index, and HbA1C), which showed good diagnostic potential for severe liver fibrosis prediction.
1. Did the authors validate the composite scoring system in a valuation cohort? The new scoring systems worked well in the current cohort in this study, if it can apply to more patients, that would be great.
2. The composite scoring system was first introduced on page 7 after describing the results in all the tables; however, the composite score was already shown in Tables 1-4. The order did not make sense. The authors can consider introducing the composite score before Tables 3 and 4 and after Tables 1 and 2. Therefore, the composite score in Tables 1 and 2 should be removed and the score in Tables 3 and 4 should be kept.
3. Please indicate in Table 4 which OR and P values are generated by univariate or multivariate analysis.
4. Lin 189 on page 4, read "The concordance between the NFS score (>0.65) and the pSWE values (>1.7m/s) was 189 met in 134 patients (88.7%)", which is confusing. Does that mean that 134 patients had both high NFS scores (>0.65) and pSWE values (>1.7 m/s) or not? Please rephrase the sentence.
5. Please include the full name of NPV, PPV, pSWE when the abbreviations were firstly described on page 1, lines 26 and 32; on page 2 line 63.
6. Typo errors: line 52 "obtain o prompt", line 187 "statically".
Author Response

(The authors gave the same response as above.)

Reviewer 5 Report
In this article, the authors focused on T2DM to explore the new protocol to predict severe liver fibrosis using various noninvasive methods. Anthropometry and serum biomarkers, such as LDL, HDL, INR, HBA1c, HOMA-IR have been collected, B-ultrasound has been used to evaluate the degree of liver steatosis, NFS score and pSWE values have been used to score liver fibrosis level. Authors found liver function tests, platelets and triglycerides didn’t associate with liver fibrosis, but GGT, HOMA-IR, HbA1c percentage, and BMI had a positive correlation with fibrosis, so a new scoring system combining these parameters had been set up, and it has higher sensitivity and specificity to predict severe liver fibrosis in T2DM accompany NAFLD patients. The authors mentioned the limitations in this article, patient case was limited, no liver biopsy to verify liver fibrosis level, the restricted accuracy of pSWE, but it is still good clinical research to explore a new noninvasive method to predict F3/F4 liver fibrosis in NAFLD based on T2DM patients.
Author Response

(The authors gave the same response as above.)

Round 2
Reviewer 1 Report
The paper has been improved. I recommend it for publication.
Author Response

(The authors gave the same response as above.)

Reviewer 3 Report
Dear Editor and dear Authors,
I found the manuscript in the present version significantly improved if compared with the previous one. However, there are still some aspects which require further revision. More specifically:
- Line 184-189: I think the rephrased sentence is not correct. Indeed, "non continuous variables" (categorical ones) are expressed as percentage, as reported in the first version of the paper. medians and IQR should be used for continuous variables with non normal distribution.
- The authors stated that "We performed the multivariate analysis using only the significant covariates in univariate analysis (AST/ALT, like other variables, had poor results in the analysis and were not presented in the table)". I think all the variables significantly different at univariate analysis shown in Table 2, should be included in table 3 and 4.
- In my opinion, the present version, discussing the composite score at the end of the result section is much better than the previous one. However, if the authors state that the composite score performed better, they should formally compare it to the other clinical variables/scores used. In particular, they have added the AUC for NFS in table 3, but they did not compare it with the AUC of the composite score.
Author Response

(The authors gave the same response as above.)
